# Thermoplastic Starch Composites Reinforced with Functionalized POSS: Fabrication, Characterization, and Evolution of Mechanical, Thermal and Biological Activities

**DOI:** 10.3390/antibiotics11101425

**Published:** 2022-10-17

**Authors:** Raja Venkatesan, Ramkumar Vanaraj, Krishnapandi Alagumalai, Shakila Parveen Asrafali, Chaitany Jayprakash Raorane, Vinit Raj, Seong-Cheol Kim

**Affiliations:** School of Chemical Engineering, Yeungnam University, Gyeongsan 38541, Korea

**Keywords:** thermoplastic starch, functionalized POSS nanoparticles, mechanical strength, antimicrobial activity, food packaging

## Abstract

Rapid advancements in materials that offer the appropriate mechanical strength, barrier, and antimicrobial activity for food packaging are still confronted with significant challenges. In this study, a modest, environmentally friendly method was used to synthesize functionalized octakis(3-chloropropyl)octasilsesquioxane [fn-POSS] nanofiller. Composite films compared to the neat thermoplastic starch (TS) film, show improved thermal and mechanical properties. Tensile strength results improved from 7.8 MPa to 28.1 MPa (TS + 5.0 wt.% fn-POSS) with fn-POSS loading (neat TS). The barrier characteristics of TS/fn-POSS composites were increased by fn-POSS by offering penetrant molecules with a twisting pathway. Also, the rates of O_2_ and H_2_O transmission were decreased by 50.0 cc/m^2^/day and 48.1 g/m^2^/day in TS/fn-POSS composites. Based on an examination of its antimicrobial activity, the fn-POSS blended TS (TSP-5.0) film exhibits a favorable zone of inhibition against the bacterial pathogenic *Staphylococcus aureus* and *Escherichia coli*. The TS/fn-POSS (TSP-5.0) film lost 78.4% of its weight after 28 days in natural soil. New plastic materials used for packaging, especially food packaging, are typically not biodegradable, so the TS composite with 5.0 wt.% fn-POSS is therefore of definite interest. The incorporation of fn-POSS with TS composites can improve their characteristics, boost the use of nanoparticles in food packaging, and promote studies on biodegradable composites.

## 1. Introduction

Soft materials based on bioplastics have been used for a long time to produce food containers, containers for storing food and packing for food products [1,2,3,4,5]. They completely inhibit the transfer of oxygen and water vapor, minimizing unintended mass transfer and food spoilage to extend the shelf life and keep the excellence of the food [6,7,8]. Several synthetic polymers have been produced and developed [9,10], and because they are not biodegradable, they are difficult to recycle. We are surrounded by many plastics in our daily lives. As a result, polymers have caused environmental problems. Materials for food packaging from petroleum have declined in popularity due to increasing levels of contamination. In this situation, biodegradable composites become more significant. In the group of sustainable polymers, starch has been regarded as one of the potential materials because of its availability, affordability, and good biocompatibility [11,12].

Two carbohydrates combine to make a semicrystalline material called starch, with several branches, and amylose, which is a linear molecule [13]. The amounts of amylose and amylopectin vary diversely. Plasticizers are extremely volatile and are limited materials that can flow among molecules using the starch polymer chains, minimizing inter- and intramolecular relationships [14].

To meet the demands for food packaging, factors such as low cost, processability, and the amount of raw material required have all been studied in the case of hydrocarbon plastics [15,16,17,18,19,20,21,22]. Most biopolymers have not been as successful as TS because of their distinctive features of availability for production and thermal stability [23,24,25]. TS biopolymers, however, pose a challenging alternative for food packaging materials due to their weak mechanical behavior and limited ability to form an effective oxygen and water barrier. However, the poor antimicrobial characteristics of TS prohibit its wide application in food packaging [26,27,28]. Thus, the main objective of TS is still to improve the mechanical, water, and oxygen barrier while maintaining its essential antimicrobial activity.

To increase the mechanical and physical intestinal barrier and antimicrobial activity, biopolymers have been combined with nanoparticles (NPs), such as inorganic metal and metal oxides, CNTs, graphene oxide, and montmorillonite [29,30,31]. Functionalized POSS nanoparticles are considered to have a strong chance of improving biopolymer/POSS composites because of their unique structure [32]. Since fn-POSS has many hydroxyl groups that interact extensively with the TS matrix by hydrogen bonds, it can diffuse and maintain itself in a reasonably well-dispersed form. This performance could be enhancing the TS/fn-POSS composite mechanical strength, thereby encouraging their implementation as new food packaging. On other hand, the essential investigations of TS/POSS composites often are addressed in previous studies [33,34].

In this work, solution casting has been used to produce TS/fn-POSS composite films, with fn-POSS synthesized under microwave-assisted conditions. To study the structure and morphologies of TS/fn-POSS composites, instruments including scanning electron microscopy (SEM) and Fourier transform infrared (FTIR) spectroscopies have been used. However, morphologies and physicochemical properties, such as mechanical, oxygen transmission rate (OTR), and water vapor transmission rate (WVTR), as well as antimicrobial activity, are strongly affected by fn-POSS contents. The clean starch film and composites’ morphological and chemical characteristics, as well as mechanical and thermal stability, have also been determined. The results indicated that TS/fn-POSS composites can be used more significantly in the food packaging sector with decreased barrier properties and increased mechanical, thermal, and antimicrobial activity.

## 2. Results and Discussion

### 2.1. Characterization of the fn-POSS Nanoparticles

FTIR spectroscopy was used to confirm the synthesis of the fn-POSS, as shown in Figure 1A. The Si-O-Si and C-O stretching vibrations are represented by the two sharpest peaks in the FTIR, which are at 1110 cm^−1^ and 970 cm^−1^, respectively. By bending CH_2_, the FTIR peak at 1487 cm^−1^ is formed. There is confirmation of the Si-O-Si, C-O vibration in the peaks at 795 and 650 cm^−1^, accordingly. The fn-POSS revealed a spectrum at 3000 cm^−1^ which was linked to the -OH groups’ distortion, perhaps as a result of the absorbed water molecules. The (101), (004), and (200) crystal planes are indicated by the diffraction peaks at 25.3°, 38.3°, and 48.0°. Figure 1B depicts the final structure. This is confirmed by the fn-3D POSS’s chemical structure, which is shown in Figure 1C. Figure 1D shows SEM images of fn-POSS. The standard size of the cubic form of fn-POSS is 50 nm. In Figure 1E,F, TEM images of fn-POSS are depicted. Strong intra- and intermolecular interaction with fn-POSS led the particles to cluster in the image, giving the appearance of them clustered. The ImageJ program was used to determine the average fn-POSS’s diameter which was around 50 nm. The long cubic shape of the fn-POSS encourages a strong tie to the matrix and provides a barrier for O_2_ and H_2_O.

### 2.2. Characterization of the TS/fn-POSS Composite Films

#### 2.2.1. Structural and Morphological Analysis

The FTIR of the TS and TS/fn-POSS composite materials are shown in Figure 2A. The starch spectrum exhibited a noteworthy and extensive absorption peak at 3320 cm^−1^, which was formed by bound C-H that was loose on the starch which is stretched, both inter- and intramolecularly. The stretching of the -CO bond led to peaks in the starch wavelength at 1183, 1090, and 922 cm^−1^, respectively. A suggestion for the peak at 1040 cm^−1^ is the Si-O-Si stretching vibration. The 1740 cm^−1^ peak relates to the stretching vibrations of C=O. The unique absorption peaks which are connected to the TS polymer units can be observed in the spectra of the TS/fn-POSS composite films. In other words, the FTIR spectra of the TS/fn-POSS composites indicate that after the combination of fn-POSS, the film remains, clearly showing the distinct and unique peaks of TS. This is provided that the FTIR bands of the TS and TS/fn-POSS composites were alike, or that there is no interaction between TS and fn-POSS or the POSS typical peak and the TS characteristic peak overlap. The structural characteristics of the composites composed of TS and TS/fn-POSS were tested using XRD. The XRD patterns of TS fn-POSS nanostructures with various weight % are represented in Figure 2B. With a high at around 2θ values of 17.4° and 28.3°, all samples exhibit a broad intensity, which is suggestive of the TS matrix’s dominant amorphous structure [35,36]. XRD studies also reveal that films containing 1 wt.% of fn-POSS are slightly crystalline. As a result of the composites’ well-miscible crystalline component, its crystallinity is less than that of the component. The TS/fn-POSS composites’ XRD peaks were invariant in comparison to the TS matrix, confirming that the addition of fn-POSS did not appear to affect the crystalline structure of TS [37,38]. The addition of fn-POSS to the TS had no noticeable impact, based on the FTIR and XRD spectra. The TEM image of a TS/fn-POSS composite 0 wt.% of fn-POSS particles is shown in Figure 2C. It suggests that the TS matrix contains certain fn-POSS aggregation. On comparing the TEM images of composite films with different fn-POSS amounts (0.5, 1 and 3 wt.%), it is apparent that a low fn-POSS content leads to a uniform dispersion of fn-POSS in TSs, and high fn-POSS content (5 wt.%) results in the presence of fn-POSS aggregates (Figure 2D). SEM was used to study the surface structure of the TS matrix and TS/fn-POSS composites.

The results of the SEM analysis are shown in Figure 3a–e. High polymer compatibility was shown by the smooth, uniform surface of the TS film in Figure 3a, which was devoid of cracks [39]. At lower amounts (0.5 and 1 wt.%), fn-POSS dispersion on the TS matrix was random and uniform, respectively, with much less aggregation (Figure 3b,c). Therefore, the tensile strength and barrier property test results were in good agreement with the results in that the uniform dispersion of fn-POSS could strengthen the barriers and mechanical properties of composites. However, at a higher loading (5 wt.%) in TSP-5.0, a substantial amount of surface aggregation of fn-POSS was found (Figure 3e). The 3 wt.% of fn-POSS film aggregate particles in the CS matrix resemble plate-like structures (Figure 3d). Among these nanocomposite films, TSP-1.0 may have the best surface shape due to its much more uniform dispersion of nano fn-POSS.

#### 2.2.2. Thermal Stability

Figure 4 shows TGA thermograms for TS and TS/fn-POSS nanocomposites, while the corresponding data are shown in Appendix A. The initial degradation of the TS and TS/fn-POSS composites from 220–350 °C may be commended for the evaporation of water and reduction of volatile compounds. With a weight loss of 39.2%, the TS matrix (TSP-0.0) showed the second stage of degradation at 382–460 °C, which was driven by the thermal degradation of the acetal groups in TS. The onset of thermal degradation of the TSP-0.5, TSP-1.0, TSP-3.0, and TSP-5.0 was further increased by the addition of 0.5, 1.0, 3.0, and 5.0 wt.% of fn-POSS at 225, 247, 275, and 350 °C, respectively. This could be a function of the composite film’s hydrogen bonds between molecules and the interactions of a coordination character between the TS’s OH groups and the methyl groups of the fn-POSS. In comparison to the TS matrix, the thermal stability of TSP-1.0, TSP-3.0, and TSP-5.0 composites was increased by about 17–29%. According to our study, the higher thermal degradation temperatures for TS/fn-POSS composites are explained due to the fn-POSS NPs’ hugely increased surface area (smaller size) as more normally found within the TS matrix. The residual weight increased as even more nanoparticles were introduced to the polymer matrix, as can be seen in the results below.

#### 2.2.3. Mechanical and Barrier Properties

The mechanical strength of the TS/fn-POSS composite is dependent on fn-POSS NPs, as seen in Figure 5a. With POSS contents, the TS/fn-POSS composite indicated a rise in tensile strength. This good interaction adhesion may be due to the physical interaction between the fn-POSS and the TS. The presence of many fn-POSS aggregates allows the tensile strength to remain nearly static as the fn-POSS contents were raised by 5%. The decrease in elongation at the break with the 5.0 wt.% fn-POSS content may be an indicator of the functionality of fn-POSS aggregation. The TS/fn-POSS composite’s O_2_ and H_2_O barrier properties, in addition to its mechanical performance, are a significant indication for food packaging materials [40]. Figure 5b shows the WVTR and OTR values for the TS and TS/fn-POSS composites. Pure TS had a WVTR value of 72.4 g/m^2^/day. With an increase in fn-POSS in the TS matrix, the WVTR values decreased. The lowest WVTR value for TSP-5.0 was 48.1 g/m^2^/day because oxygen molecules find it difficult to permeate the TS matrix when fn-POSS NPs are present; one of the most essential components of packaging material is its oxygen permeability. The results of measuring the OTR values of TS/fn-POSS nanocomposites are shown in Figure 5b. Data are shown in Appendix A. Because of the presence of the fn-POSS content, the value of OTR ranges from 51.8 to 140.2 cc/m^2^/24 h. The films’ barrier characteristics were enhanced by adding fn-POSS to the TS matrix. The interaction of two phenomena produced diffusion through the polymers: The area allowed for gas diffusion decreased as fn-POSS NPs were loaded, and the distance that gaseous molecules must travel from across films increased. It was found that the OTR values of the nanocomposites were lower than the TS matrix. The addition of the fn-POSS did not specifically improve the crystallinity of the TS in this investigation. The higher tortuosity of the oxygen molecule’s diffusion path, given by the evenly distributed fn-POSS, is thought to be the reason for the nanocomposites’ lower OTR (0.5, 1.0, 3.0, and 5.0 wt.%). When fn-POSS is added, the OTR value of the nanocomposite decreased by up to 5.0 wt.%. Similar reports have been published about the other nanoparticle/polymer systems as well [41], where oxygen or vapor are prohibited from passing through nanoparticles, thereby increasing the shelf life of food.

#### 2.2.4. Water Contact Angle Analysis (WCA)

To study the surface characteristics of TS/fn-POSS composites and identify the effects of the filler content, wettability measurements, or WCA, were performed. Figure 6 shows the water contact angle values and measurements taken with distilled water as the probing liquid. The hydrophobic characteristic of neat TS is testified to by its contact angle of 56.4°. This increased significantly with increased fn-POSS content, showing the presence of the filler at the film-air interaction and that the hydrophobic characteristics of the material have also enhanced (because there are propyl and silica groups from the filler). The WCA value of the TS/fn-POSS (TSP-5) nanocomposite film is 76.3° (Oliveira et al. have observed similar results) [42].

#### 2.2.5. Antimicrobial Activity of TS/fn-POSS Composite Films

Over recent years, composites based on nanoparticles have become increasingly used in biological applications for their antimicrobial activity [43,44]. The essential feature of food packaging materials is antimicrobial activity. Using the zone of inhibition, as shown in Figure 7a, *S. aureus* and *E. coli* were used as a test group for the antimicrobial activity of TS and TS/fn-POSS composites. The results are represented in Appendix A. On the clean TS (TSP-0.0), there is no inhibition zone against *S. aureus* and *E. coli*. The TS/fn-POSS composites, in contrast, show an obvious antimicrobial zone of inhibition. The diameters of the antimicrobial zones in the TS/fn-POSS composites are depicted in Figure 7b. The powerful antimicrobial activity of the (TSP-5.0) composites against *S. aureus* and *E. coli* suggests that the addition of fn-POSS can improve antimicrobial activity. Additional confirmation that the TS/fn-POSS composite film effects are real and significant against *S. aureus* microorganisms is provided by the somewhat smaller antimicrobial zone widths for *S. aureus* greater in comparison to those for *E. coli* and *S. aureus*. This was tested to confirm the zone of inhibition activity of TS in the TS/fn-POSS composite, and the results were similar to all those stated in other reports [45]. According to studies [46,47], the TS biodegradable polymer has strong antimicrobial activities against gram-positive and gram-negative microorganisms. TS-based biodegradable plastics with some constituents have a maximal contribution and effective antimicrobial activity [48,49].

#### 2.2.6. Biodegradability of TS and TS/fn-POSS Composite Films

Biodegradable packaging films fabricated from renewable materials are required to preserve food quality, be long-lasting, and be less wasteful in their manufacture [50]. Changes in the weight of the TS/fn-POSS films were used to assess their biodegradation. The biodegradability levels of the TS/fn-POSS composite films were evaluated using the weight loss of specimens at 7, 14, 21, and 28 days, as shown in Figure 8. The extent of the material deterioration largely depends on the chemical structure and humidity levels. The TS/fn-POSS composite film (TSP-5.0), with a hydrophilic nature, demonstrated the highest % of weight loss after 28 days. The soil moisture could quickly absorb the polymer network, shortening the polymer chains and raising the likelihood of soil microorganisms breaking down the amylopectin. TSP-3.0 only degraded to about 50% of its initial state after 28 days which was the lowest degradation of all the chemically synthesized films. This indicates that biodegradation was not a simple path for the chemical interactions in the polymer. It is interesting to note that the TSP-3.0 and TSP-5.0 films, which included 3.0 and 5.0 wt.% of fn-POSS, respectively, biodegraded as fast as TS film (around 62.5 and 78.4%, respectively, in 28 days). Due to their mechanical strength and intermolecular interactions, composites and mixed materials are stronger than the TS matrix alone. However, TS seems to be less vulnerable to soil microorganisms [51,52,53]. These results demonstrate that the degradation rates of the TSP-3.0 and TSP-5.0 were greatly affected by the addition of fn-POSS [54].

## 3. Materials and Methods

### 3.1. Chemical and Materials

The thermoplastic starch in this study was obtained from S.D. Fine Chemicals in India. The majority (97%) of the obtained 3-chloropropyl trimethoxysilane was obtained from Alfa Aesar in India. Acetic acid, dichloromethane, and methanol (99%) were purchased from Sigma Aldrich in India. All chemicals obtained were used without any further purification.

### 3.2. Synthesis of fn-POSS Nanoparticles

As per earlier research [55], the characterization and properties of the synthesized fn-POSS were studied. Figure 9 shows that octakis (3-chloropropyl)silsesquioxane was formed (fn-POSS). The solution was prepared by combining 5 mL of concentrated HCl with 150 mL of methanol, and then transferred to a two-necked round-bottomed flask with a condenser and Teflon-coated magnetic pellets. The solution was stirred at 25 to 30 °C for 30 min. Thirty grams (0.075 mol) of (3-chloropropyl) trimethoxysilane was added continuously to the solution through a funnel for 45 min while the mixture was thoroughly stirred for approximately two hours. By allowing the mixture to sit at 25 to 30 °C for 48 h without stirring, the methoxy group was fully dissolved. Then, 0.30 g, 0.024 mol of di-n-butyltin dilaurate was introduced as a catalyst to aid the Si-OH groups self-condense. The reaction mixture formed a white solid precipitate after being maintained correctly for 2 days. Before being dried in an air-circulating oven to obtain fn-POSS, it was filtered, times washed three times with methanol to increase purity, and afterward subjected to FTIR analysis. FTIR analysis resulted in the following: 1635 (cm^−1^) (C-C stretching), 1110 (cm^−1^) (Si-O-Si stretching vibrations), 970 (cm^−1^) (C-O stretching vibrations), 795, and 650 (cm^−1^) (bending vibrations of Si-O-Si). The characterization information agrees with the previous article [56].

### 3.3. Preparation of TPS/fn-POSS Nanocomposites

The TS/fn-POSS composites were produced by solution mixing and drop-casting, as shown in Figure 10. First, TS was dissolved in hot water. After being dissolved in the same solution with the required amount of fn-POSS for 12 h, the suspension was processed ultrasonically for 30 min at room temperature. The TS solution and fn-POSS NPs suspension were left constantly swirling for 12 h before the solution was sonicated for a further 30 min [57]. After cooling to room temperature, the mixture was poured onto a clean glass petri dish. Recent suggestions for how to clean using the Piranha solution are presented [58]. The TS/fn-POSS composites were produced after the solvent had disappeared. After the films were dried in a vacuum oven at 60 °C for 24 h to remove some of the remaining solvent residues, the dry composites were recovered by peeling them off the petri dish. For the sake of clarity, the pure TS and TS/fn-POSS composite film was called TSP-0.0, TSP-0.5, TSP-1.0, TSP-3.0, and TSP-5.0, accordingly.

### 3.4. Characterization

#### 3.4.1. Fourier Transform Infrared Spectroscopy (FTIR)

An FTIR spectrophotometer was used to determine the film sample’s FTIR spectrum at a resolution of 4 cm^−1^ (Perkin Elmer spectrophotometer RX1). The 2 × 2 cm squares of the prepared specimens were instantly sliced and placed in the sample holder. The wavenumber band for the spectra was 4000–400 cm^−1^.

#### 3.4.2. X-ray Diffraction Analysis (XRD)

The composites were tested by XRD at room temperature using an analytical diffraction meter (Rigaku, Mini Flex 120 II-C, Tokyo, Japan). After the specimens were prepared according to the instructions and deposited (2 × 2 cm) on a glass plate, the spectra were collected.

#### 3.4.3. Morphological Analysis

With the help of a microscope with a 5.0 kV accelerating voltage, the TS/fn-POSS and fn-POSS NPs were micrographed by SEM. The TS/fn-POSS composite samples were distributed on a carbon tape surface. Prior to imaging, a gold sputter coating was applied to every sample. The TEM test was carried out using an electron microscope with a 200 kV acceleration voltage (FEI TECNAI T-20).

#### 3.4.4. Thermogravimetric Analysis (TGA)

To evaluate the thermal stability, a thermogravimetric tester (QA 50, TA Instruments, New Castle, DE, USA) was used. All samples were heated at a rate of 10 °C/min to temperatures ranging from 25 to 700 °C in a nitrogen atmosphere.

#### 3.4.5. Mechanical Strength

A universal testing machine (H10KS, Tinius Olsen, Salfords Surrey, UK) was used to test the tensile strength of materials with the following aspects: 25 mm wide, 50 mm in length, and 0.085 mm thick (ASTM-D882-02). After the examination of five samples, the average value was calculated.

#### 3.4.6. Oxygen Transmission Rate (OTR)

An oxygen permeability tester was used to evaluate the OTR of the TS/fn-POSS composites at 25 °C (Noselabats, Bovisio-Masciago, Italy). The composites were measured several times in various locations to get at the average value. All tests were performed in daylight.

#### 3.4.7. Water Vapor Transmission Rate (WVTR)

The WVTRs of the fn-POSS nanoparticles combined with TS were investigated and measured using a water vapor permeability analyzer (Mocon Permatran, Minneapolis, MN, USA). All water vapor permeability tests were performed at room temperature in accordance with the ASTM F-1249 standards.

#### 3.4.8. Water Contact Angle Analysis

The water contact angle of the TS/fn-POSS composites was evaluated using a goniometer (Holmarc, Kochi Kerala, India) and the sessile drop method at 23 °C and 50% RH. To determine the contact angle, a 1 L droplet of water was dropped onto the coated surface, and an image of the droplet was recorded within 5 s. The mean value was calculated as the average of the contact angle measurements made at five different locations on the film. The experiment’s error rate was 1°.

#### 3.4.9. Antimicrobial Activity

The method of disk diffusion was used to investigate the antimicrobial activity of the TS matrix and its composites [59,60]. Two food-based pathogenic bacteria, *E. coli* and *S. aureus*, were used to assess the antimicrobial activities of the films in their as-prepared state. Professor Jintae Lee of Yeungnam University in the Republic of Korea donated the films on the *E. coli* and *S. aureus* strains (MSSA, *E. coli* O157:H7 EDL933) for in vitro studies. The films were originally purchased from the American Type Culture Collection (ATCC). The culture media, microbial inoculum size, and incubation conditions were in accordance with the CLSI’s recommendations [61]. After the plates were incubated for two days at 37 °C, the plates were examined for clear zones.

#### 3.4.10. Biodegradability Tests

The evaluation of biodegradation was performed using a method that has been described in the literature [62,63,64]. TS/fn-POSS composites were used to make samples of size 2.5 × 5.0 cm, which were then weighed and buried in a thick layer of decomposing material. For optimum microbiological activity, water was constantly added to the compost every day. The samples were removed from the vacuum oven after three standard intervals of 0, 7, 14, and 21 days, washed with tap water and dried for 12 h to remove any moisture. Based on the following Equation (1), the weight loss was calculated.
(1)$$W_{\mathrm{loss}}\left(\%\right)=\frac{W_{o}-W_{t}}{W_{o}}\times 100$$
where W_o_ and W_t_ represent the pre- and post-burial weights of the specimens in the compost soil. According to the average results from up to five specimens for each of the fn-POSS contents, the final results were reported.

#### 3.4.11. Statistical Analysis

To show repeatability, all studies were conducted five times. The mean and standard deviation of all the statistical data were calculated using SPSS 22.0. To determine the major differences between groups, a one-way analysis of variance (ANOVA) was used. For statistical significance, a difference had to be below 0.05. SigmaPlot was used to present the results.

## 4. Conclusions

In the current study, solution casting was employed to produce TS/fn-POSS composite films. The FTIR results showed that there was intermolecular hydrogen bonding between the TS matrix and fn-POSS. The uniform and homogeneous dispersion of fn-POSS on the TS matrix in TEM and SEM micrographs was confirmed by TSP-3.0. The elongation at break and tensile strength of the TSP-5.0 film increases due to the addition of fn-POSS to the TS matrix, raising its mechanical strength and allowing it to be studied as a structurally stable food packaging film. The WVTR and OTR for TS/fn-POSS composites were substantially lower compared to TS. By enhancing the water contact angle, fn-POSS also produced TS materials which were more hydrophobic. By increasing the amount of fn-POSS, the TS’s water contact angle improved from 56.4° to 76.3° which enhanced hydrophobicity. The TS/fn-POSS composite film displayed an impressive antimicrobial action against microorganisms, which increased with increasing fn-POSS (*S. aureus* and *E. coli*). Therefore, the TSP-5.0 film may be proposed as a suitable organic fruit and vegetable packing film. The results indicate that TS/fn-POSS composites can be utilized as materials for food packaging which minimize the microbiological load and extend the shelf life of packaged foods.

## Figures and Tables

**Figure 1 antibiotics-11-01425-f001:**
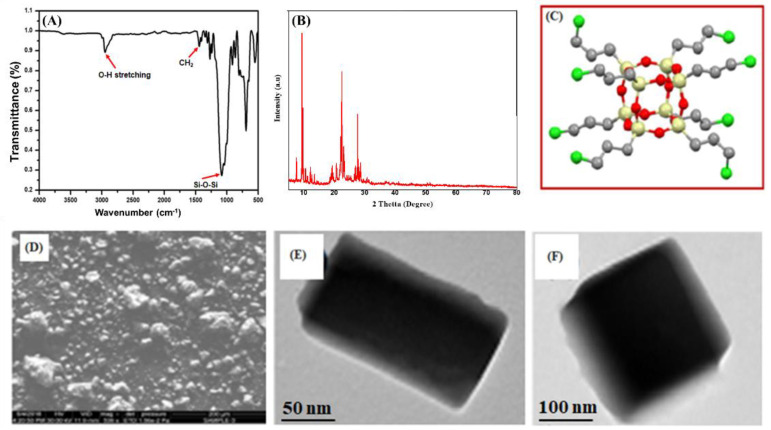
Characterization of fn-POSS structural and morphological characteristics: (**A**) FTIR; (**B**) XRD analysis; (**C**) 3D Chemical structures of fn-POSS; (**D**) SEM image of fn-POSS; (**E**,**F**) TEM images of fn-POSS at different magnifications (with the addition of a drop of dichloromethane solution, the sample was deposited on a copper grid that had been coated with carbon, and was then air-dried).

**Figure 2 antibiotics-11-01425-f002:**
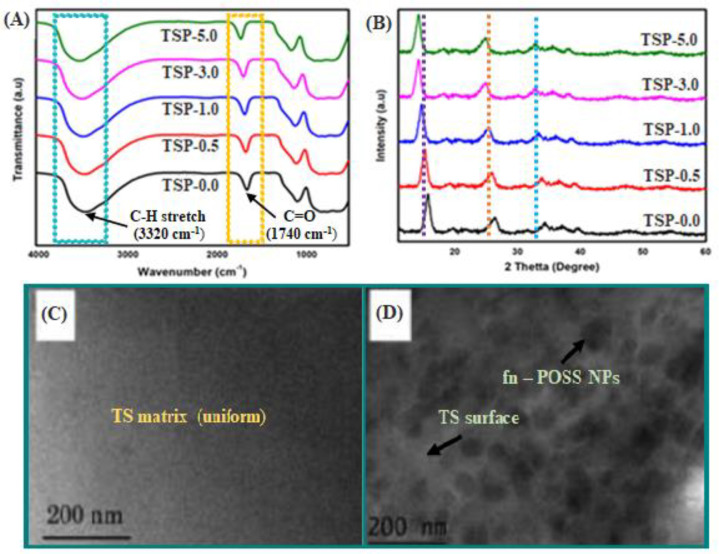
Structural and morphological characterizations of TS/fn-POSS nanocomposite films with different amounts of fn-POSS: (**A**) FTIR spectra; (**B**) XRD patterns, (**C**,**D**) TEM images. The fn-POSS content of nanocomposite is 0.0 and 3.0 wt.% for the TEM images.

**Figure 3 antibiotics-11-01425-f003:**
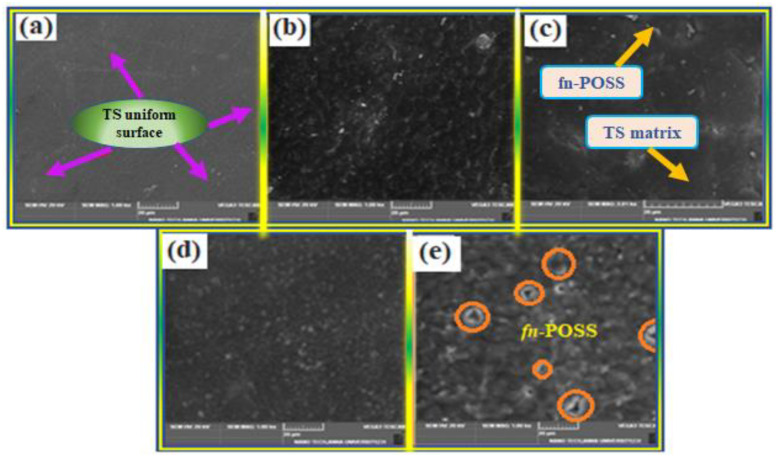
SEM images of TS matrix and its composites; (**a**) 0.0, (**b**) 0.5, (**c**) 1.0, (**d**) 3.0 and (**e**) 5.0 wt.% of fn-POSS nanoparticles.

**Figure 4 antibiotics-11-01425-f004:**
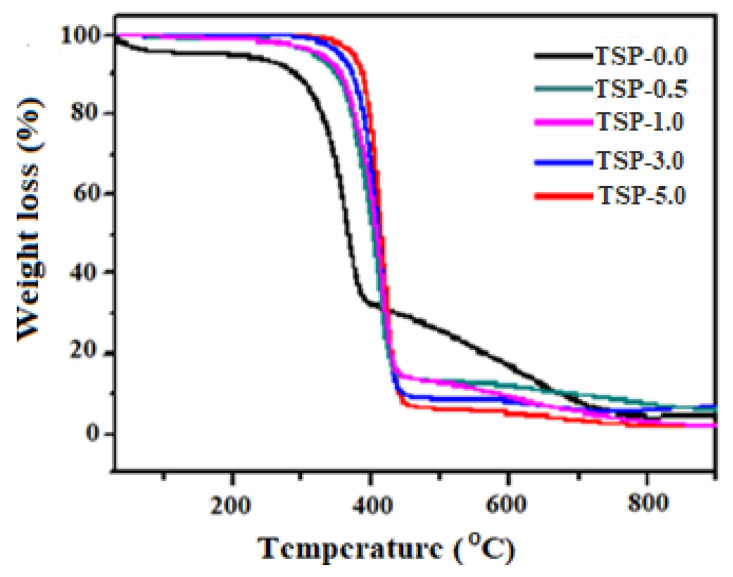
TGA curves of TS and TS/fn-POSS nanocomposite samples.

**Figure 5 antibiotics-11-01425-f005:**
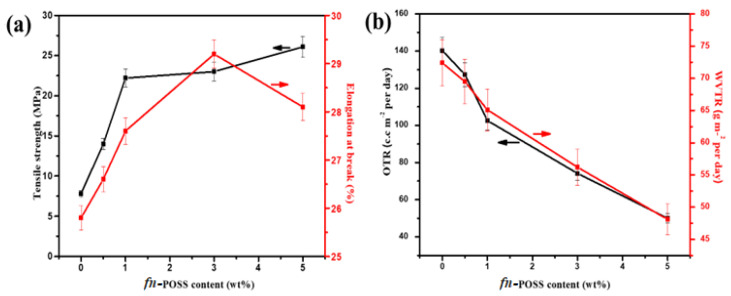
(**a**) Tensile strength and Elongation at the break vs. fn-POSS content; (**b**) OTR and WVTR values vs. fn-POSS content.

**Figure 6 antibiotics-11-01425-f006:**
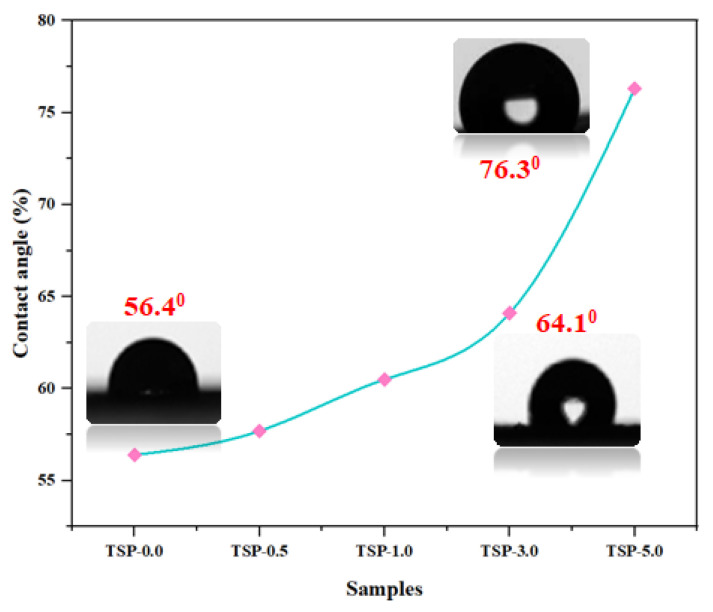
Water contact angle measurements for TS and TS/fn-POSS nanocomposites with a different weight percentage of filler (fn-POSS).

**Figure 7 antibiotics-11-01425-f007:**
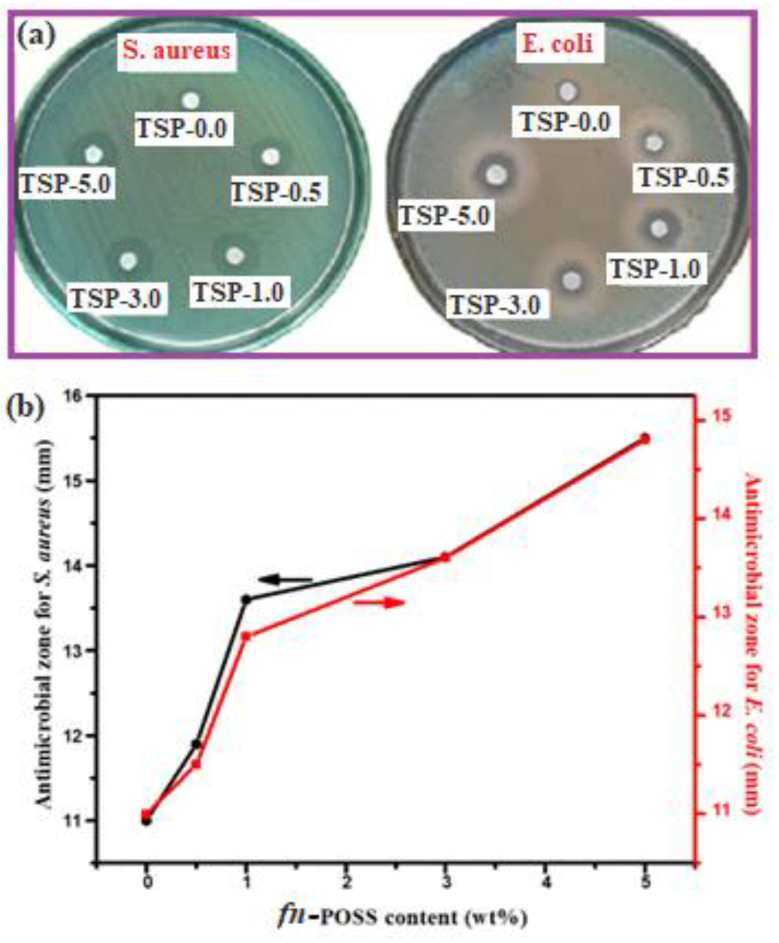
(**a**) Antimicrobial activities of TS and TS/fn-POSS composite samples against *S. aureus*, and *E. coli*; (**b**) Antimicrobial zone of inhibition diameter vs. fn-POSS content.

**Figure 8 antibiotics-11-01425-f008:**
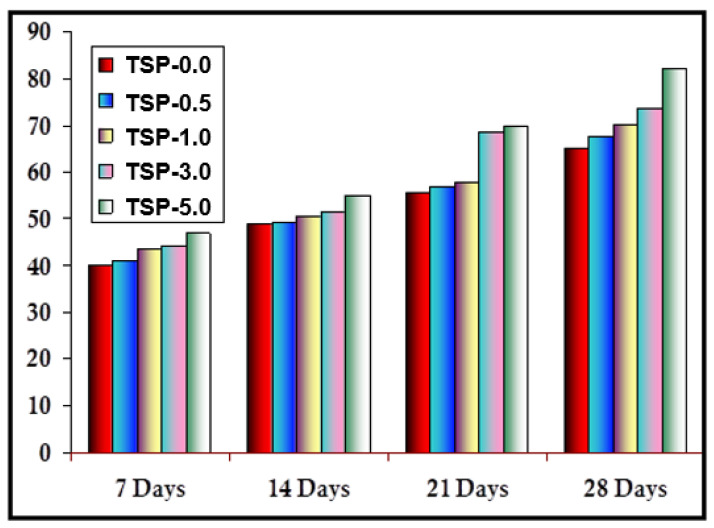
Soil biodegradability of TS, and TS/fn-POSS at different intervals of days.

**Figure 9 antibiotics-11-01425-f009:**
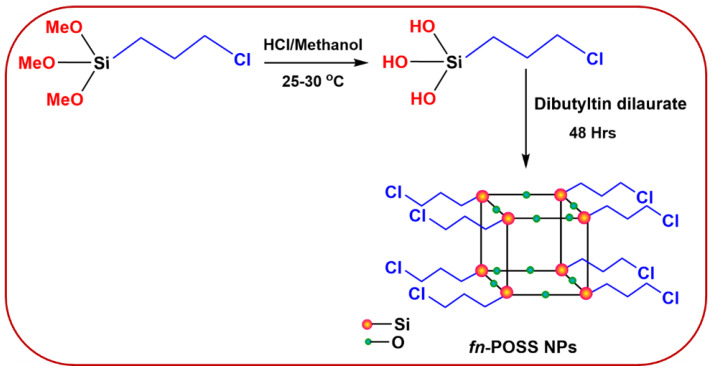
Synthesis of octakis(3-chloropropyl)silsesquioxane.

**Figure 10 antibiotics-11-01425-f010:**
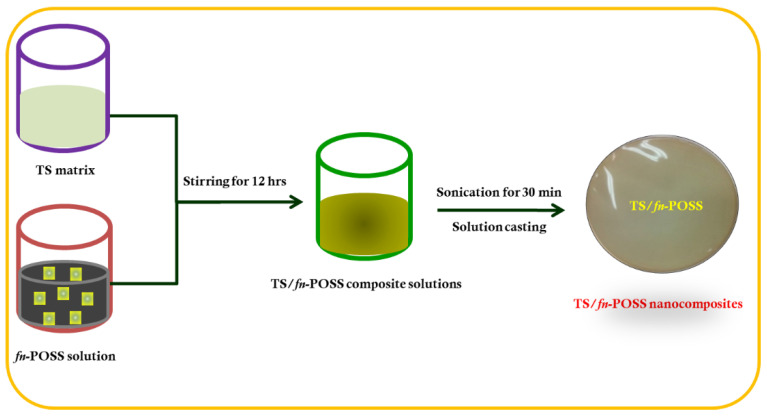
Fabrication of TS/fn-POSS nanocomposites (0.0 to 5.0 wt.% of fn-POSS NPs).

## Data Availability

Not applicable.

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
