# Peer review of "Thermoplastic Starch Composites Reinforced with Functionalized POSS: Fabrication, Characterization, and Evolution of Mechanical, Thermal and Biological Activities"

_antibiotics, 2022, doi:10.3390/antibiotics11101425_

Round 1

Reviewer 1 Report

Over all plagiarism checking is needed.

Introduction to be modified 

Reviewer 2 Report

Dear Authors,

There are some issues that should be checked before the manuscript has been accepted.

For (2.2.5). There is no discussion about the antimicrobial activitity of the similar nanobiocomposites. However, there are so many publications about nanocomposite bioactive/biodegradable food packaging materails in the literature. 

For (2.2.6) Same as above, there is no discussion part about the biodegradability of the sample.

For (3.4.9) Did the authors use any reference for antimicrobial activity? where did authors obtain the strains?  How did they prepared the inoculums? 

For (3.4.10) Same as above, there is no reference about the biodegradability of the sample.

For (3.4.11) Did the authors repeat the antimicrobial activity test for five times? If so, they should give the standard deviation values for inhibition zone measurements (Table S3).

Overall, the discussion is not adequate throughout the manuscript.

Reviewer 3 Report

Dear Authors,

The manuscript is well written, and has scientific significance. The experimental design is well planned and presented. Numerous techniques have been used so as to study the mechanical, thermal and antimicrobial properties of thermoplastic starch composites reinforced with functionalization of POSS. From the derived results the improvement of the characteristics due to the addition of fn-POSS to TS composites is clearly defined. The explanation and discussion made by the Authors is well presented. The conclusions are supported from the data presented and are meaningful. Format, grammar and units are correct.

Round 2

Reviewer 2 Report

Check the uploaded file. There are too many English mistakes. 
